# OpenReview forum: "JailNewsBench: Multi-Lingual and Regional Benchmark for Fake News Generation under Jailbreak Attacks"
_ICLR.cc/2026/Conference — ICLR 2026 Poster_

### Official Review · Reviewer_kfeM · 2025-10-29

**Soundness:** 3
**Presentation:** 3
**Contribution:** 3
**Rating:** 6
**Confidence:** 3

**Summary:**

This paper introduces JailNewsBench, a large-scale benchmark for evaluating LLM robustness against jailbreak-induced fake news generation. The benchmark spans 34 regions and 22 languages, incorporating 8 evaluation sub-metrics and 5 jailbreak attacks, totaling around 300k instances. The authors report that the maximum attack success rate reached 86.3%, with notably higher success and generation quality for English and U.S.-related topics, underscoring the need for multilingual and region-aware evaluation.

**Strengths:**

(1) This paper proposes the first multilingual and regional benchmark specifically for jailbreak-induced fake news generation. It covers an impressive scope: 34 regions, 22 languages, 5 attack types, 8 sub-metrics, and ~300k instances.

(2) This paper provides comprehensive evaluation across 9 state-of-the-art LLMs, revealing meaningful cross-lingual and regional disparities.

(3) This paper presents an evaluation showing strong correlation between LLM-as-a-Judge scores and human ratings

**Weaknesses:**

(1) Region selection excludes politically sensitive or unstable regions, skewing the dataset toward developed areas and limiting representativeness.

(2) Evaluation relies on GPT-5, Claude, and Gemini—the same families under test—introducing potential model-family bias and circular evaluation.

(3) It is unclear whether evaluation was performed in each native language or after translation; translation-based judging may distort non-English results.

**Questions:**

(1) Please clarify the two-stage filtering process—what exactly is excluded or retained, and how many samples fall into each category?

(2) How is Attack Success (AS) computed—before or after filtering?

(3) Were judgments made in the original languages or English translations? If translated, how was translation quality verified?

(4) Why do English and U.S. topics show higher attack success and quality?

---

> ### Author Response · Authors · 2025-11-21
>
> We sincerely appreciate your valuable and constructive comments.
> We have carefully addressed each of your suggestions, and the corresponding revisions are highlighted in **green** in the manuscript.
> Please find our detailed responses below.
>
> ## Weaknesses
>
> - *Politically sensitive or unstable regions were excluded during dataset collection:*
>     - Thank you for your comment. We respectfully note that this is not a weakness but rather a deliberate ethical safeguard. As stated in Section 2.1, the exclusion of politically unstable regions is an intentional decision to mitigate potential harm in case our fake news generation methods are misused. From an ethical and safety perspective, we consider it our responsibility as researchers to ensure that harm prevention takes precedence, and we stand by this design choice as responsible research practice.
> - *Bias from LLM-as-a-Judge:*
>     - Thank you for raising this concern. We conducted additional experiments in Appendix I to investigate potential bias by comparing LLM-as-a-Judge scores when the evaluation model and generation model are the same versus different. The results show no significant difference between these conditions. Furthermore, as demonstrated in Section 3, our LLM-as-a-Judge framework achieves high correlation with human judgments in meta-evaluation. Therefore, we can conclude that LLM-as-a-Judge bias and circular evaluation do not occur in our setting. (Note: This overlaps with another reviewer's concern, so the text is highlighted in purple.)
> - *Distortion of meaning when meta-evaluations are translated into English:*
>     - Thank you for seeking this clarification. As stated in Section 2.4 on meta-evaluation, "Three annotators per language performed the annotation," meaning that native speakers evaluated the outputs in their respective languages. Therefore, there is no noise from translation-based evaluation in our results.
>
> ## Questions
>
> - *Number of Instances Excluded through Two-Stage Filtering:*
>     - Thank you for this question. In Table 2, IFL and AS correspond to the first-stage and second-stage evaluations, respectively, and indicate the proportion of data to which each was applied. This shows how much of the data was captured in each stage of the evaluation.
> - *When AS Is Calculated:*
>     - AS is computed before any filtering is applied, as stated in Section 2.4.
> - *English and U.S. topics show higher attack success and quality:*
>     - This is the opposite: the English-speaking regions or the United States show higher attack success rates and greater harmfulness in the generated news compared to other regions. This may be because safety measures are primarily centered around English-speaking or U.S.-based regions [1].
>         - [1] Wenxuan Wang, Zhaopeng Tu, Chang Chen, Youliang Yuan, Jen-tse Huang, Wenxiang Jiao, Michael R. Lyu. All Languages Matter: On the Multilingual Safety of Large Language Models. ACL 2024.

---

> > ### Comment · Reviewer_kfeM · 2025-11-26
> >
> > Thanks for the clarification. Some concerns are not fully addressed:
> >
> > (1) The explanation for why English/U.S. topics show higher attack success and quality remains unclear and internally inconsistent. The manuscript reports lower AS for English/U.S. regions, whereas the rebuttal states the opposite. This discrepancy needs to be reconciled, and a consistent, evidence-based explanation provided.
> >
> > (2) Thank you for clarifying the ethical rationale behind excluding politically sensitive or unstable regions. However, my original concern remains: this choice substantially skews the dataset toward developed, politically stable regions, limiting representativeness. Even if ethically motivated, the resulting geographic imbalance should be acknowledged as a methodological limitation, and claims about broad regional generalizability should be calibrated accordingly.

---

> > > ### Author Response · Authors · 2025-11-26
> > >
> > > - Thank you for pointing this out. The inconsistency you noted stems from a typographical error in our rebuttal text, not from the manuscript itself. As you correctly stated, “*the defense capabilities are higher in the English/U.S. regions*,” and this is the intended and correct description. This point is consistently presented in the current manuscript. For example, Figure 1 shows that the defense capabilities in the English/U.S. regions are higher than in other regions, and our explanation is based on these results. To prevent any possible confusion, we have refined the wording in the revised manuscript to more clearly and consistently convey this point.
> > > - Thank you for your constructive feedback. While we had already noted this point as a limitation in the main text, we agree that its ethical implications merit clearer articulation. Following your suggestion, we have expanded the discussion within the *Ethical Considerations* section to explicitly address your point. At the same time, we would like to clarify that **the primary objective of our dataset is not global representativeness**. Rather, our aim is to construct region-specific datasets and to evaluate defensive capabilities within the ethically and legally permissible scope. From this perspective, **the exclusion of certain regions does not compromise the methodological limitation of our study**; instead, it reflects necessary ethical and legal constraints inherent to general research design in this field.
> > >
> > > We hope that these revisions adequately address all your concerns!

---

### Official Review · Reviewer_eRvd · 2025-10-31

**Soundness:** 2
**Presentation:** 2
**Contribution:** 2
**Rating:** 4
**Confidence:** 3

**Summary:**

This paper proposes JailNewsBench, a benchmark designed to evaluate the vulnerability of large language models (LLMs) to jailbreak attacks in the context of fake news generation. The authors introduce a systematic framework that includes: A set of seven attack strategies (Original, Explicit, Role Play, System Override, Research Front, Negative Prompting, Context Overload) to elicit harmful fake news from LLMs.An LLM-as-a-Judge evaluation framework with eight sub-metrics to assess the harmfulness of generated fake news.Experiments across multiple languages and models to demonstrate the effectiveness of the proposed benchmark.

**Strengths:**

<1> A large-scale, multilingual, multi-region benchmark specifically targeting jailbreak-induced fake news generation, covering diverse geographies/languages, seed rationales, and attack strategies, and supporting both black-box and white-box evaluations.

<2> An interpretable scoring framework with eight sub-metrics on a 5-point scale that moves beyond a single aggregate score; it improves reliability and diagnostic power, and is validated against alternative judges/human annotations.

<3> Systematic evaluation across strong LLMs uncovers key safety weaknesses, language/region disparities, and gaps in current safety benchmarks, offering actionable insights for model alignment and defense design.

**Weaknesses:**

<1> The relationships among sub-metrics and their aggregation method require further clarification.
The paper proposes eight evaluation dimensions, but does not provide detailed explanations regarding the independence among these metrics, how weights are assigned, or how the final overall score is computed. Some metrics may be highly correlated (e.g., "Verifiability" and "Faithfulness"), which could affect the efficiency and interpretability of the evaluation.

<2> The credibility of the evaluation results needs to be further strengthened.
The current assessment relies entirely on  models such as GPT-5, Gemini 2.5, and Claude 4 as judges. Although these models are highly capable, their judgments may contain implicit biases or suffer from insufficient consistency. It is recommended to supplement the evaluation with a human annotation experiment to analyze the agreement between human annotators and LLM-based judges.

<3> The presentation details could be further improved.
 The logical flow between certain sentences could be smoother. Additionally, the lack of sufficient figures, tables, and illustrative examples affects the overall readability.

<4>Insufficient discussion of failure cases or edge cases
The paper primarily presents successful jailbreak examples, but lacks analysis of instances where the model successfully resists attacks.

<5>The issue of redundancy among sub-metrics
Regarding Appendix F, Table 8 shows the top three sub-metrics with the highest average scores for each model. It can be observed that most models include Formality, Adherence, and Faithfulness. This raises the question: are the eight metrics potentially redundant?

**Questions:**

Q1. How do you ensure sub-metric independence and a robust aggregation into the final score?
Please (a) report pairwise correlations (Pearson/Spearman) among the 8 sub-metrics and run VIF, factor analysis or PCA; (b) state the exact weighting/aggregation formula and how weights are chosen (fixed vs. learned; per-dataset or global); (c) conduct sensitivity/ablation by removing or reweighting highly correlated metrics (e.g., Verifiability, Faithfulness, Formality, Adherence) and show whether model rankings remain stable (Kendall’s τ / Spearman ρ with CIs). Directly address the redundancy suggested by Appendix F, Table 8.

Q2. How reliable are the LLM-judge scores without human raters?
Please add a human annotation study and report agreement with LLM judges (Cohen’s κ / Krippendorff’s α / Spearman ρ), plus inter-judge consistency across GPT-5, Gemini 2.5, and Claude 4 (variance, CIs via bootstrap). Provide the judging prompts and decoding settings (temperature, seed) and show that conclusions hold under alternative judges/ensembles or majority-vote calibration.

Q3. What are the failure and edge cases, and can presentation be strengthened with concrete examples/plots?
Please include paired examples where attacks fail (i.e., the model resists) alongside successful jailbreaks: prompt - model response - per-sub-metric scores - final decision. Summarize error/defense taxonomies by attack type/task. Improve readability with figures/tables (e.g., radar plots of 8 metrics, box/violin plots of variance, ranking-stability plots) and illustrative case studies that clarify the logical flow.

---

> ### Author Response · Authors · 2025-11-21
>
> We sincerely appreciate your valuable and constructive comments.
> We have carefully addressed each of your suggestions, and the corresponding revisions are highlighted in **pink** in the manuscript.
> Please find our detailed responses below.
>
> ## Weaknesses
>
> - *Redundancy of submetrics:*
>    - Thank you for this important feedback. We would like to clarify that your concerns can be addressed.
>    - **Sub-metric design and validation:** We manually designed the sub-metrics with reference to existing discussion on harmful fake news characteristics, carefully considering their independence, importance, and coverage of diverse aspects.
>        - **Independence verification:** We confirmed that the correlation coefficients among most sub-metrics remain less than 0.25 and VIF score is low, demonstrating that **the sub-metrics independently evaluate different aspects of fake news.** (We have added this discussion on the relationships among sub-metrics to Appendix G.)
>         - **Ablation study:** We conducted ablation experiments on the sub-metrics (Table 12 in Appendix G). The results show that while the three sub-metrics ranked highest in Table 11 (originally Table 8) exhibited particularly large performance drops when removed, all sub-metrics led to significant performance degradation when ablated, **confirming the importance of each sub-metric.**
>     - Aggregation method: We use a simple average of sub-metrics for the final score. While weighting schemes would be an effective approach from a performance optimization perspective, they require preparing language-specific datasets and tuning the parameters using them, and most importantly, **our research objective is to provide a generalizable cross-lingual evaluation framework for fake news generation**. Therefore, we experimentally demonstrate that the sub-metrics are complementary to each other and that simple averaging enables robust performance across diverse languages. We have added this to Section 2.
>     - Furthermore, we conducted additional experiments demonstrating that sufficient performance can be achieved both with direct weighting and simple averaging. We added the results and discussion of this analysis, in which we explored the weights of each sub-metric using four-fold cross-validation on the meta-evaluation dataset, to Appendix J. Table 15 presents the differences in meta-evaluation correlations between region-specific weighted sub-metrics and averaged sub-metrics, as well as the variance of the optimized weights. The experimental results show that less than half of the cases exhibited statistically significant differences, and the variances were small. This finding aligns with the results from the ablation and correlation analyses, supporting the view that each sub-metric evaluates distinct aspects and that their diversity contributes to overall performance.
>    - We believe these findings and explanations directly address your concern.
> - *Correlation between LLM-as-a-Judge and human evaluations:*
>     - Thank you for this important concern. As described in Section 3 (Meta-Evaluation of LLM-as-a-Judge), we have already conducted human annotation experiments to validate our LLM-based evaluation. We confirmed that our LLM-as-a-Judge approach achieves significantly higher correlation with human annotations compared to existing evaluation metrics and single-score baselines. This result demonstrates the reliability of our LLM-based evaluation framework.
> - *Improvement of logical flow and presentation of concrete examples:*
>     - Thank you for this feedback. We have revised the paper to improve the logical flow between sections. Regarding the lack of concrete examples of generated fake news, from a safety perspective, we deliberately do not include directly usable LLM-generated fake news examples in the paper to prevent immediate misuse. This strategy follows the guidelines from Ovadya and Whittlestone (2019).
>     - We have added an explanation for this decision in the Ethical Considerations section.
> - *Examples of successful model defenses against attacks:*
>     - Thank you for this feedback. As mentioned earlier regarding safety considerations, we do not include concrete examples of fake news generation in the paper. However, we have added a detailed analysis in Appendix F (Table 10) examining which parts of the prompt contribute to harmful fake news generation. (Note: This experiment answers Reviewer kiio's comment as well, and thus the corresponding text is highlighted in blue.)
>     - Our analysis reveals that jailbreak instructions contribute most to attack success. Therefore, **models with lower attribution to jailbreak prompts can be considered more successful at resisting attacks**.
>
> ## Questions
>
> - All the Questions were related to the Weaknesses, so we addressed them all in the Weaknesses section.

---

> > ### Comment · Reviewer_eRvd · 2025-11-28
> > **Thank you for your response. It has resolved many of my doubts and concerns.**
> >
> > After reading the authors’ response and the revised manuscript, I believe several of my main concerns have been substantially alleviated.
> >     I hope the authors will revise the final version of the manuscript to clearly address these issues, and I am inclined to recommend acceptance of this paper.

---

> > > ### Author Response · Authors · 2025-11-28
> > >
> > > Thank you very much for your thoughtful follow-up.
> > >
> > > - Among the issues you initially raised, we believe the only point we could not directly address in the manuscript concerns the inclusion of LLM-generated fake news examples. As we hope is now clear, this was not due to a methodological or technical limitation of our study, but rather a deliberate ethical precaution. We will also ensure that the final version of the manuscript fully and faithfully incorporates all revisions and clarifications included in the current revised version.
> > > - If you find our responses satisfactory, we would be grateful if you could reflect this assessment in your final score as well.

---

### Official Review · Reviewer_kiio · 2025-11-01

**Soundness:** 2
**Presentation:** 3
**Contribution:** 2
**Rating:** 2
**Confidence:** 4

**Summary:**

This paper introduces JailNewsBench, a large-scale benchmark for testing whether jailbreak attacks can induce LLMs to generate fake news across 34 regions and 22 languages (≈300k seed instructions). Seed instructions are grounded in real news and four malicious “motivations”, then paired with five jailbreak styles (i.e., role-play, system override, research front, negative prompting, context overload). Outputs are scored by an LLM-as-a-Judge that decomposes harmfulness into eight sub-metrics (e.g., Faithfulness, Subjectivity, Agitativeness). On nine LLMs, the authors report high attack success and non-trivial harmfulness, with English/U.S. topics generally more robust than other languages/regions; translating non-English items into English does not close the gap. Models only modestly detect their own generated fakes (best F1 is only 68.2). The authors also argue fake-news generation is underrepresented in existing safety datasets relative to toxicity/bias.

**Strengths:**

* Clear societal risk focus, broad coverage. The benchmark targets jailbreak-induced fake-news generation, spanning 34 regions/22 languages and ~300k instances, which is substantially broader than typical English-centric setups.

* Documented selection guardrails. The paper explicitly specifies region inclusion criteria (exclude places with special fake-news laws, high instability, or very recent news) to reduce release risk.

**Weaknesses:**

LLMs’ vulnerability in misinformation-related contexts is a well-established finding. This paper revisits the issue through two main empirical observations: (1) LLMs can be induced to generate misinformation through prompting, and (2) they struggle to detect LLM-generated fake news when relying solely on their internal knowledge and without access to external evidence. However, both observations have been widely reported in prior work: [1] involves the first point, [1-3] involves the second point.

Consequently, the paper’s primary novelty appears to rest on its scale, multilingual scope, and the design of its jailbreaking prompts. Nonetheless, several aspects limit the strength of this contribution:

* Lack of comparative validation. Although the paper positions its jailbreak design as a key contribution, there is no systematic validation of its superior effectiveness, e.g., whether it achieves higher success rates than existing studies that prompts LLMs to generate misinformation [1], produces more harmful or influential misinformation, or yields more deceptive outputs that better evade detection.

* Absence of per-type analysis. The study defines five jailbreak strategies and two baselines, but the results aggregate all into a single “Jailbreak” condition. A per-type comparison would clarify which jailbreaks are most effective, and under what conditions, providing more insight into their distinct mechanisms.

* Under-theorized model compliance. While the paper introduces diverse jailbreak patterns, it does not explore why models comply with deceptive instructions. Analyses on refusal bypassing, instruction-following cues, or safety-policy framing could substantially strengthen interpretability. Similarly, ablations linking prompt design to compliance behavior are missing. This reduces the utility of this work in informing research on LLM safety.

* Focus on attack efficacy, not governance. The paper systematically explores how to increase jailbreak success but offers no parallel study of defenses or governance (e.g., retrieval grounding, output filtering, or post-hoc detection pipelines) nor an evaluation of how these mitigations affect success/harmfulness. As a result, the work advances offense without producing actionable risk-reduction guidance for practitioners.

* This manuscript raises additional ethical concerns. Please refer to Flag For Ethics Review.

[1] Can LLM-Generated Misinformation Be Detected? ICLR 2024.

[2] Bad Actor, Good Advisor: Exploring the Role of Large Language Models in Fake News Detection. AAAI 2024.

[3] Fake News in Sheep's Clothing: Robust Fake News Detection Against LLM-Empowered Style Attacks. KDD 2024.

**Questions:**

Please address all weaknesses and ethical concerns raised.

**Details Of Ethics Concerns:**

* Human subjects without ethics details. In Lines 131–132, the authors state: “We outsourced all translation tasks and human evaluations by native speakers through Upwork.” As human subjects were hired and exposed to misinformation (their awareness on the information's veracity remains unclear), this likely requires IRB/ethics approval and clear documentation of annotator instructions, consent, compensation, and safety protocols. The paper provides no explicit information on any of these.

* Controlled access claim vs. public release of jailbreak prompts. In Line 053, the paper says the benchmark will be “released post-acceptance with controlled access”. However, Appendix B and Appendix C already publicly released prompt templates that elicit fake-news generation with demonstrated high success rates, which materially lowers the barrier to abuse and could enable rapid, multilingual disinformation at scale.

* Missing Ethics Statement. Given the sensitivity of the topic and the risks above, it is fundamentally inappropriate that the paper lacks a dedicated Ethics Statement addressing these issues.

---

> ### Author Response · Authors · 2025-11-21
>
> We sincerely appreciate your valuable and constructive comments.
> We have carefully addressed each of your suggestions, and the corresponding revisions are highlighted in **blue** in the manuscript.
> Please find our detailed responses below.
>
> ## Weaknesses
>
> - *Detection and Defense Strategies for Fake News*
>     - Thank you for your valuable feedback.
>     - We respectfully note that the novelty concern raised in W1 does not constitute a weakness of our work, as our primary contributions are the first multi-lingual, multi-regional fake news dataset construction, proposing a fine-grained evaluation framework, and proposing a jailbreak setting tailored to fake news generation.
>     - Nevertheless, we acknowledge that conducting additional experiments about internal representation strengthens our study, which also directly addresses W2.
>     - Specifically, we conducted additional experiments investigating whether fake and real news can be classified by analyzing the model's hidden layer representations, inspired by prior work showing that internal representations often contain information about hallucination and counterfactuals not reflected in surface-level outputs.
>     - **Our key finding**: Linear probes on hidden layers achieve significantly higher classification accuracy than relying on the model's output.
>     - **Perspectives defense (For W2)**: **This finding has direct implications for practical defense strategies,** specifically, it suggests that monitoring internal representations during inference can enable more effective output filtering and post-hoc detection pipelines than approaches based solely on analyzing generated text.
>     - **Additional novelty regarding** LLMs’ vulnerability (for W1): **Our work is the first to demonstrate classification feasibility using internal representations**.
>     - We believe this addresses your concerns, and have added this discussion to Section 3.
> - *Comparison with existing prompts that make the model generate fake news:*
>     - Thank you for this valuable suggestion to strengthen our work.
>     - We have added the experimental results against the existing prompting approach from [1] by applying their prompts to our dataset. The results are presented in Table 8.
>     - **Our key findings**: Our approach consistently achieves significantly higher attack success rates and LLM-as-a-Judge scores compared to [1]. This demonstrates that our method more effectively elicits harmful misinformation from LLMs.
>     - We have added this discussion to Section 3.
> - *Performance by type of jailbreak:*
>     - Thank you for your feedback. The two baselines are presented separately in Table 2 with slash-separated values, but for the five jailbreak strategies, as you mentioned, we initially presented their average to show overall trends.
>     - Following your suggestion, we have now added per-type results for all five jailbreak strategies in Table 9.
>     - Key findings:
>         - System override tends to elicit the most harmful fake news across most models.
>         - For context overload, while it achieves high attack success rates, it receives lower LLM-as-a-Judge scores. This indicates that although it successfully induces models to generate fake news, the quality tends to be lower.
>     - These insights provide a clearer understanding of the distinct mechanisms and effectiveness of different jailbreak strategies. We have added this discussion to Appendix E.
> - *Analysis of model instruction following:*
>     - Thank you for this insightful suggestion to deepen our analysis.
>     - To investigate why models comply with deceptive instructions, we conducted a gradient-based attribution analysis to identify which parts of the input prompt contribute most to harmful fake news generation. The results are presented in Table 10.
>     - Key findings: Models with higher attribution to jailbreak instructions achieve higher AS, while models with higher sub-metric scores show greater attribution to the original instructions. This reveals that jailbreak instructions bypass safety mechanisms, while content quality depends on detailed guidelines for generation.
>     - We have added this discussion to Appendix G.

---

> ### Author Response · Authors · 2025-11-21
>
> ## Details Of Ethics Concerns
>
> - *Details of the Ethical Aspects of Crowdsourcing:*
>     - Thank you for raising this important concern. While we had included information about annotator compensation in Appendix A, you are correct that insufficient ethical details were provided in the paper. We have now added a dedicated **Ethical Considerations** section that includes detailed annotation instructions and all relevant information. The information previously in Appendix A has been consolidated into this new section.
>     - **Key clarifications:**
>         - This crowdsourcing study was approved by our institutional ethics board.
>         - **For translators:** No fake news content was provided. They were only given prompt templates to translate, not any misinformation.
>         - **For meta-evaluation annotators:** They were explicitly informed that the content was AI-generated fake news and were asked to evaluate its quality with this knowledge.
>         - **For all workers:** We disclosed our institutional affiliation, our roles, and the purpose of the research (studying LLM-generated fake news). All workers provided informed consent for data usage.
>         - **Compensation:** All workers were paid $20 USD per hour.
> - *Release of Prompts and Dataset:*
>     - Thank you for this important concern. We carefully considered the mitigation of abuse risks for each component we release, following the framework of "deployment barriers" and "ease of deployment for mal-use" from Ovadya and Whittlestone (2019).
>     - **Our risk-based release strategy:**
>         - **Appendix B & C (prompt templates):** These are templates that require additional resources (e.g., dataset construction) to be operationalized, thus maintaining higher deployment barriers. Therefore, we release them for transparency and defense development.
>         - **Benchmark prompts:** These are ready-to-use with no technical expertise required, representing low deployment barriers. Therefore, we apply controlled access to minimize immediate misuse risk.
>     - Also, controlled release of attack information benefits the research community by (1) enabling defense mechanism development and detection system improvements, and (2) allowing reproducible evaluations under the same threat model, thereby advancing safety measures.
>     - As additional safeguards in our benchmark design, we only target historical news (dated between August 8, 2020 and November 29, 2022) as stated in Section 2.1. We also exclude politically unstable regions identified in the Fragile States Index and Conflict Watchlist 2025 to reduce risks of exacerbating anxiety or serious consequences.
>     - **Our risk-based release strategy enables us to share valuable insights from a defensive perspective while minimizing the risk of our research being misused.** We have added this discussion to the Ethical Considerations section.
> - *Missing Ethics Statement:*
>     - Thank you for highlighting this. While we had discussed ethical issues in the main content, such as Section 2, and thus did not include a separate Ethics Statement, following your feedback, we have now added a dedicated Ethics Statement section to clearly articulate the ethical aspects of our research.

---

### Official Review · Reviewer_KVsi · 2025-11-02

**Soundness:** 3
**Presentation:** 3
**Contribution:** 3
**Rating:** 6
**Confidence:** 4

**Summary:**

This paper evaluates the robustness of LLMs against jailbreak-induced fake news generation. It introduces JailNewsBench, the first large-scale multilingual and regional benchmark for this task. The benchmark spans 34 regions and 22 languages, incorporates five types of jailbreak attacks, and assesses model outputs using an eight-dimensional LLM-as-a-Judge harmfulness evaluation framework. Experiments on nine prominent LLMs reveal high attack success rates (up to 86.3%) and significant disparities in robustness between English and non-English regions. The study highlights fake news generation as an underexplored yet critical dimension of LLM safety, complementing existing work on toxicity and bias.

**Strengths:**

1. This paper systematically studies jailbreak attacks that induce LLMs to generate fake news, addressing an important and underexplored safety gap in current research.
2. The proposed JailNewsBench spans multiple regions and languages, making it comprehensive and representative than existing jailbreak benchmarks.
3. The authors conduct extensive experiments across different LLMs, including both black-box models and open-source ones, using four malicious motivations, five jailbreak techniques, and an eight-dimensional harmfulness evaluation. The paper provides detailed descriptions and appendices that should facilitate reproducibility.

**Weaknesses:**

1. The paper does not provide a grouped analysis of attack performance under different malicious motivations. It would be interesting to see whether current LLMs exhibit varying sensitivity to these different intent types.
2. The assessment relies on LLMs to evaluate outputs, which may introduce circular or model-specific bias.
3. Considering that fake news generation is a highly sensitive and potentially harmful topic, the paper would benefit from a more detailed discussion on ethical safeguards and dataset release protocols.
4. While the study identifies vulnerabilities, it does not explore mitigation strategies or future directions such as defense mechanisms or fine-tuning approaches to resist jailbreak attacks.
5. Minor typos, e.g.,  Appendix B title: “PROMPT FOR GENERATING EED INSTRUCTIONS.”

**Questions:**

What is the specific criterion or threshold of “successful” attacks?

**Details Of Ethics Concerns:**

The paper introduces methods and datasets that demonstrate how existing LLMs can be exploited to generate fake news. While the intent is to evaluate safety vulnerabilities, such techniques and data could potentially be misused for harmful purposes. Therefore, the authors should include a more thorough ethical discussion and ensure strict safeguards for dataset access and responsible use.

---

> ### Author Response · Authors · 2025-11-21
>
> We sincerely appreciate your valuable and constructive comments.
> We have carefully addressed each of your suggestions, and the corresponding revisions are highlighted in **purple** in the manuscript.
> Please find our detailed responses below.
>
> ## Weaknesses
> - *Attack performance under different malicious motivations:*
>     - Thank you for pointing this out. We have added the experimental results for each malicious motivation to Appendix H. We confirmed that the attack with "political" motivation achieved the highest attack success rate, leading to more harmful fake news generation. These findings indicate that models exhibit different vulnerabilities depending on the underlying motivation, highlighting the importance of motivation-aware defense strategies.
> - *Bias from LLM-as-a-Judge:*
>     - Thank you for your comment. We have added an analysis investigating the difference in sub-metrics scores between cases where the models used for evaluation and fake news generation are the same versus different (Appendix I, Table 14). We found no significant difference between them, indicating that such bias does not occur, at least in our LLM-as-a-judge evaluation.
> - *To safely release the dataset*:
>     - Please refer to our response in the "Details of Ethics Concerns."
> - *Consideration of defense strategies:*
>     - Thank you for raising this important point. To explore potential mitigation strategies, we have added new analysis of classification accuracy between fake news and original (faithful) news using linear probes on LLMs' hidden layers (Section 3, Table 4). Our results show that using internal representations achieves significantly higher classification accuracy than relying solely on output text. This finding suggests a promising direction for defense mechanisms, such as output filtering and fake news detection pipelines that monitor internal model states. We have added this discussion.
> - *Minor typo:*
>     - Thank you for the comment. We have fixed the typo.
>
> ## Questions
>
> - *Definition of attack success rate:*
>     - As described in Section 2.4, we provide the LLM-as-a-Judge model with the question “Does this output refrain from refusing to answer?”, and determine attack success based on whether the model’s output avoids refusal. This evaluation criterion is widely adopted in many jailbreak studies [1,2].
>         - [1] Patrick Chao, Edoardo Debenedetti, Alexander Robey, Maksym Andriushchenko, Francesco Croce, Vikash Sehwag, Edgar Dobriban, Nicolas Flammarion, George J. Pappas, Florian Tramer, Hamed Hassani, Eric Wong. JailbreakBench: An Open Robustness Benchmark for Jailbreaking Large Language Models. NeurIPS 2024.
>         - [2] Tinghao Xie, Xiangyu Qi, Yi Zeng, Yangsibo Huang, Udari Madhushani Sehwag, Kaixuan Huang, Luxi He, Boyi Wei, Dacheng Li, Ying Sheng, Ruoxi Jia, Bo Li, Kai Li, Danqi Chen, Peter Henderson, Prateek Mittal. SORRY-Bench: Systematically Evaluating Large Language Model Safety Refusal. ICLR 2025.
>
> ## Details Of Ethics Concerns
>
> - *To safely release the dataset*:
>     - Thank you for your comment. As you pointed out, measures to prevent misuse are necessary for our dataset. We have adopted a risk-based release strategy following Ovadya and Whittlestone (2019).
>     - **Our release strategy:**
>         - **Templates (Appendix B & C):** Publicly available, as they require additional resources (e.g., dataset construction) for actual misuse, maintaining higher deployment barriers.
>         - **Benchmark prompts:** Controlled access, as they can be immediately used by anyone regardless of technical expertise, representing low deployment barriers.
>         - **Original news content**: We only target historical news (dated between August 8, 2020 and November 29, 2022)
>     - **Additional safeguards:**
>         - We exclude politically unstable regions identified in the Fragile States Index and Conflict Watchlist 2025 to reduce risks of exacerbating serious consequences.
>     - These measures minimize misuse risks while enabling valuable insights for defense research. We have added this discussion to the Ethical Considerations section (highlighted in blue, as this section also addresses other reviewers' comments).

---

### Author Response · Authors · 2025-11-26

Dear Reviewers,

Again, thank you very much for your constructive feedback.

This is a gentle reminder that we have posted a brief response on OpenReview and uploaded the revised manuscript.
To facilitate your review, we have used different font colors to distinguish the revisions made in response to each reviewer, and we hope this helps clarify which changes correspond to which reviewer’s comments.

If you have any additional questions or require further clarification, we are ready to respond immediately.

We hope that our revisions will help you to positively re-evaluate our work.

---

### Author Response · Authors · 2025-12-04

Dear Reviewers, AC, SAC, and PCs,

Thank you for engaging with our work. While not all reviewers participated in the discussion phase, we appreciate the positive responses from those who did.

**All reviewers praised our multilingual, multicultural jailbreak-based fake news benchmark** (including attack design, dataset construction, evaluation protocols, and human validation), recognizing that both fake news and jailbreaks are critical safety challenges. They acknowledged our work plays an important role in LLM safety, a significant recognition for our Datasets and Benchmarks submission.

We addressed all concerns through nine additional experiments and a revised manuscript:
- (i) **Ethics clarified**: Added a dedicated Ethical Considerations section (building on details already in the main text) covering IRB approval, crowdsourcing, and staged data/prompt release. **Regional restrictions and phased data release reflect standard ethical/legal constraints in this field, not methodological limitations**.
- (ii) **Probing reveals hidden robustness**: Linear probing on internal representations detects fake news more effectively than zero-shot prediction, a novel finding suggesting LLMs are more robust than previously thought.
- (iii) **LLM-as-a-Judge validated**: Matched vs. unmatched judge-target comparisons show no significant bias. Combined with our already demonstrated strong human correlation (exceeding prior work), **this confirms the reliability of our evaluation**.

We respectfully request that you consider these strengths in your final decision.

Sincerely,

The Authors

---

### Meta-Review · Area_Chair_DMcp · 2026-01-08

**Summary:**

Reviewers broadly agree that JailNewsBench is a timely and important datasets-and-benchmarks contribution, providing the first large-scale multilingual and regional benchmark for jailbreak-induced fake news generation, with substantial scale (34 regions, 22 languages, ~300k instances), diverse jailbreak attacks, fine-grained evaluation metrics, and careful analysis revealing significant cross-lingual and regional safety disparities.

**Reviewer Concerns:**

Initial concerns focused on ethics (risk of misuse, human subjects, release protocols), evaluation validity (LLM-as-a-Judge bias, metric redundancy), novelty beyond scale, lack of per-attack/per-motivation analysis, and limited discussion of defenses; these were largely addressed in the rebuttal through a dedicated ethics section with IRB details and risk-based release, added per-motivation and per-jailbreak analyses, human–LLM correlation studies, metric correlation/ablation analyses, comparative prompting baselines, probing-based defense insights, clarified regional findings, and acknowledgment of representativeness limits due to ethical constraints.

**Reviewer Scores:**

KVsi likely remains 6 (core questions addressed, ethics clarified); eRvd explicitly indicated a shift toward acceptance 4→6; kfeM likely stays 6 after clarification of inconsistencies and limitations; overall sentiment trends upward after revisions and ethics triage confirmed no ethics review needed.

kiio hadn't given feedbacks but would plausibly move 2 to 5 or higher (many novelty, analysis, and ethics concerns directly addressed, though still skeptical about offensive focus)

---

### Decision · Program_Chairs · 2026-01-26

Accept (Poster)